# Achilles Tendon Repair after Tenorraphy Imaging and the Doughnut Metaphor

**DOI:** 10.3390/ijerph20115985

**Published:** 2023-05-29

**Authors:** Gian Nicola Bisciotti, Andrea Bisciotti, Alessio Auci, Alessandro Bisciotti, Cristiano Eirale, Alessandro Corsini, Piero Volpi

**Affiliations:** 1Paris Saint Germain Football Club (France), Kinemove Rehabilitation Centers, 54027 Pontremoli, Italy; bisciotti@libero.it; 2Kinemove Rehabilitation Centers, 54027 Pontremoli, Italy; andrea.bisciotti1@gmail.com; 3Azienda USL Toscana Nord-Ovest, 54100 Massa, Italy; alessioauci@gmail.com; 4IRCCS Humanitas Research Hospital, 20089 Rozzano, Italyvolpi.piero@libero.it (P.V.); 5Paris Saint Germain Football Club, 78100 Saint Germain en Laye, France; cristiano.eirale@hotmail.com; 6Genoa Cricket and Football Club, 20155 Genoa, Italy

**Keywords:** tendon biology, regeneration, imaging, return to play

## Abstract

After Achilles tendon tenorraphy, tendon tissue undergoes a long period of biological healing. During this period, tissue turnover shows heterogeneity between its peripheral and central regions. This case report concerns the description of the tendon healing process of an athlete who underwent an Achilles tendon tenorraphy. As the reparative processes progressed, magnetic resonance imaging (MRI) showed centralization of the hyperintensity area and the tendon assumed a doughnut-like appearance. At the same time, ultrasound (US) assessment showed a progressive reorganization of the tendon fibrillar structure. Therefore, for the athlete, MRI and US assessment together represent a useful tool for the decision-making process after Achilles tendon tenorraphy.

## 1. Introduction

Rupture of the Achilles tendon is one of the most impacting injuries in sports pathology [1] and represents the most common type of injurious tendon rupture [2]. Furthermore, this specific type of injury has increased in recent decades [3,4]. Most patients suffer an Achilles tendon rupture during a sporting activity [1,5]. Despite this, to date, there is no real consensus regarding the management of Achilles tendon ruptures [1]. The conservative management can achieve good results, but the reinjury rate may well be unbefitting for athletes [3]. Open or percutaneous surgery repair are the most common options for Achilles tendon ruptures [3]. Clinical examination should be followed by imaging, which has an important role in the decision-making process [3,6]. Indeed, in the post-surgical period, the clinician faces the problems of when to introduce active motion or the typical exercises of the patient’s sport and finally when to allow the return to play [6]. The aim of this study is to describe the repair processes of the Achilles tendon, in a professional footballer, after tenorraphy. The rehabilitation plan was monitored simultaneously with magnetic resonance imaging (MRI) and ultrasound (US) assessment.

Normal tendon tissue, in MRI spin-echo and gradient-echo sequences with common echo times > 10 ms, has a very low signal intensity [7]. This can be explained by the typically long T1 and short T2 relaxation times of the tendon, in which the partially positive extremities of the dipolar water molecules are closely associated with the collagen matrix [8,9]. Indeed, the T2 relaxation time for a non-pathologic tendon tissue is characteristically very short (i.e., on average 0.25 ms) [10] and is independent of the field strength used [11]. For this reason, tendon pathology assessment is based on the detection of the areas of hyperintensity signal within the tendon. Indeed, the presence of hyperintensity areas is due to T2 relaxation time prolongation associated with the disruption, disorganization and edema of the collagen tissue [12,13]. Gradient echo and STIR (Short-Tau Inversion Recovery) sequences are more sensitive for the detection of focal sign changes in tendon tissue [14,15]. The Achilles tendon is surrounded by soft tissue rich in fat. For this reason, fat suppression sequences increase the MRI sensitivity for tendon lesions [16,17,18,19]. Another imaging technique in tendon pathology is US examination [20]. US assessment has been used for assessing Achilles tendon rupture since the early 1980s [21]. Furthermore, it has been used in post-operative assessments of Achilles tendon reconstruction since 1990 [22,23]. A peculiar feature of US examination is the possibility of assessing tendon echotexture. In the US scan, human tendon echotexture appears as an internal network of tendinous, fine, parallel and linear echoes similar to a fibrillar structure. US assessment using 15 MHz probes is able to provide an optimal image of the inner tendon structure. Indeed, a high axial resolution allows separation of echoes from adjacent fibrils, giving a clear delineation of the fibril interspace [24,25]. Upon US assessment, a post-surgical Achilles tendon is characterized by hypoechoic thickening and a loss of normal fibrillar echotexture [20]. It is important to note that MRI scans show the tendon as a homogeneous structure and do not reveal the fibrillar structure. For this reason, US and MR imaging together prove to be paramount for an in-depth study of the biological processes of tendon repair [24,25]. Tissue turnover in the Achilles tendon is a very slow process that can be defined as “bradytrophic” [26]. Furthermore, several studies show that the core of the Achilles tendon is formed before the age of 20, and is not renewed substantially thereafter [26,27,28]. Indeed, some studies based on the ^14^C (Carbon-14, or radiocarbon) concentrations in this tendon show that the tendon core is formed during height growth and is essentially not renewed thereafter [27,29,30]. On the other hand, the tendon’s peripheral zone can respond to mechanical forces by altering its biological structure and, consequently, its composition and mechanical properties [26,27,28]. Therefore, the repair processes of the tendon are characterized by two factors. The first is the slowness of the process as a whole. The second is the fact that the repair takes place in a centripetal direction (i.e., from the peripheral tissue towards the central zone) [26,27,28]. This concept may be explained by the “doughnut metaphor”. In this metaphor, the doughnut ring is the Achilles tendon’s peripheral zone formed by aligned fibrillar structures, and the doughnut hole is the tendon’s central area of disorganization [31]. Therefore, after tenorraphy, the Achilles tendon will show, during the reparative continuum, a progressive reduction in and centralization of the fibrillar disorganization area. At the same time, the tendon shows a centripetal increase in the peripheral area consisting of healthy tendon tissue (Figure 1). The MRI assessment provides a complete view of both the peripheral area of healthy tissue (area of hypointensity signal) and the disorganization area (zone of hyperintensity signal). However, the MRI does not allow for a comprehensive evaluation of the reparative processes that take place within the disorganization area. Indeed, the MRIs show a hyperintensity signal zone as a homogeneous area, which prevents observation of the fibrillar structure. For these reasons, the concurrent study of the processes of reduction in and centralization of the disorganization area (via MRI), and of the progress of the fibrillar reorganization processes within the latter (via US), allows for an in-depth overview of the processes of biological recovery in the tendon.

## 2. Case Presentation

This case report refers to a 27-year-old professional football player belonging to the Italian first division (heigt 180 cm, body weight 89 kg). The subject underwent tenorraphy of the left Achilles tendon via the open surgical technique following a traumatic rupture during competition. The subject had never complained of tendinopathy at the level of the Achilles tendon. In the period before the tendon injury, the subject performed his training plan normally and participated in competitions. The injury occurred in December, i.e., about 6 months after the start of the season. During the entire rehabilitation period, the patient underwent four MRI assessments and the same number of US assessments. MRI and US assessments were performed simultaneously. Two different expert radiologists evaluated the MRI images in a blind manner. Likewise, the same two experienced radiologists carried out the US assessments in a blind manner. The inter-observer reliability values concerning MRI and US assessment were, respectively, equal to 0.89 and 0.85. The evolution of the MRI and US images is shown in Figure 2, Figure 3, Figure 4 and Figure 5. A summary graph of the main data from the MRI and US examinations is shown in Figure 6. It is worthy of note that the player recuperated full functionality, assessed by the single leg heel raise test [6], VISA-A score (Victoria Institute of Sport Assessment—Achilles) [32] and rebound jump test [33], and was able to return to normal training seven months after surgery. At that stage, the MRI showed a centralization of the hyperintensity area (i.e., a doughnut-like appearance). At 3 years follow-up, the player returned to the same pre-injury performance level and there were no complications or reinjuries. It is important to note that the player recuperated full functionality despite the MRI images not being completely normalized. This may indicate that the persistence of a centralized zone of hyperintensity signal does not represent an obstacle to the recovery of full functionality of the muscle–tendon unit [26,27,28] or a risk factor for the recurrence of tendon rupture [34,35].

## 3. Discussion

After surgical repair of the Achilles tendon, the average period for return to sporting activity varies from 6.5 [35,36] to 9.1 months [35,37]. During this period, the MRI images show (i) a progressive increase in healthy peripheral tissue; (ii) a progressive reduction in the hyperintensity signal area corresponding to the disorganized tendon tissue; (iii) a progressive centralization of the hyperintensity signal area; and (iv) a progressive reduction in the tendon antero-posterior aspect. This confirms the heterogeneity in tendon tissue turnover between the outer and central regions of the tendon [27,34,35]. In fact, the Achilles tendon has a negligible tendon core turnover, yet the peripheral zone of the Achilles tendon shows good biological adaptability [27,34,35]. The biological adaptability of the peripheral portion of the Achilles tendon was demonstrated by several studies based both on retained isotopic ^14^C levels [27] and on stable isotope levels and microdialysis [38,39,40]. Furthermore, this biological adaptability is also confirmed by the tendon hypertrophy induced by long-term training [6,41,42]. The presence of the area of hyperintensity signal in fluid-sensible sequences (T2 and STIR) after Achilles tendon tenorraphy is a long-term phenomenon. Indeed, 19.3 ± 11.3 months after Achilles tendon tenorraphy, in 30.7% of patients there is still a hyperintensity area present equal to 9.5 ± 10.2% of the tendon CSA [34]. The long-term persistence of this hyperintensity zone is proof both of the bradytrophic nature and the heterogeneity of tissue turnover in the Achilles tendon [26,27]. However, it is worth noting that Karjalainen et al. [34] reported that, once the area of signal hyperintensity was reduced and centralized, cases of unsatisfactory recovery from muscle–tendon unit functionality fell to 15% and no cases of re-injury occurred. In other words, once a doughnut-like appearance is achieved, the risk of re-injury is drastically reduced. This could be a clinical situation similar to that incurred after an indirect muscle injury. Indeed, following an indirect muscle injury, MRI signals normalize after a relatively long period of time, typically 6 months on average [43,44,45]. However, despite the persistence of MRI signal alteration, the percentage of re-injuries was less than 2% in all the studies considered. The presence of abnormalities in MRI signals during this period may be explained by the greater number of ionic interactions of immature collagen formed during the early stage of muscle healing. The conversion of these weaker bonds to stronger covalent bonds, during post-translational modifications of the constituent amino acids, may require longer periods. This period may be of up to 6 months depending on the extent of the injury [43]. Furthermore, Slavotinek [46] showed that a decrease of at least 70% in the area or volume of hyperintensity compared to the baseline, before return to play, is not associated with re-injury. For these reasons, the return to play decision-making process, after muscle injury, does not necessarily require total normalization of the MRI signal [43,44,45]. Therefore, the hypothesis that return to play following Achilles tenorraphy is not necessarily linked to the normalization of MRI signals may be plausible. Nonetheless, as in the case of muscle injury, a minimum value of reduction in the hyperintensity signal area in relationship to tendon CSA, below which the risk of reinjury is minimal, must be identified. Another important point to underline is the complementarity of MRI and US imaging following Achilles tendon tenorraphy. Indeed, after surgery, one of the main problems is being able to evaluate the quality of tendon tissue during the process of healing [47]. US examination can detect tendon fibril orientation and diameter, and the hyperechogenic and hypoechogenic zones associated with post-surgery tendon tissue proliferation [34,48], On the contrary, MRI imaging shows the tendon as a homogeneous structure and is unable to reveal the fibrillar structure. For this reason, the consultation of both US and MRI imaging allows an in-depth assessment of the biological processes of tendon repair [24,25].

## 4. Limitation of the Study and Future Implications

This study is limited because it is a clinical case without a sample size. Indeed, no scientific conclusion can be reached without a clinical trial. Therefore, future randomized clinical trials are needed to reach definitive scientific conclusions concerning the topic presented and discussed in this case report.

## 5. Conclusions

For both clinicians and radiologists, it is very important to recognize and distinguish images of normal and complication-related late post-operative biological changes following the surgical repair of an Achilles tendon. In the specific case of the athlete, the MRI and US assessment may represent a useful tool during the return-to-play decision-making process, both for the choice of the optimal time for resuming sporting activity and for avoiding the risk of reinjury.

## Figures and Tables

**Figure 1 ijerph-20-05985-f001:**
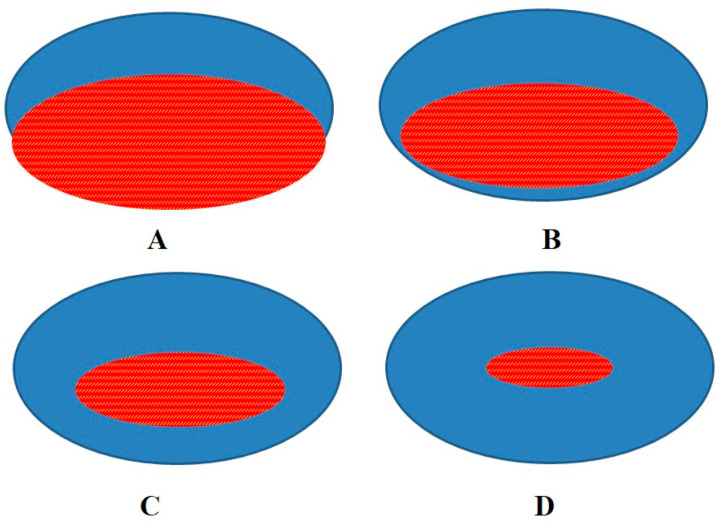
Scheme of the reparative “continuum” of the Achilles tendon after tenorraphy. The red area represents the zone of disorganization of the tendon after tenorraphy (**A**). The blue area is the zone of the tendon formed by aligned fibrillar structure (**A**). According to the progression of biological repair processes, the disorganization area becomes smaller and tends to centralize (**B**–**D**). At the same time, the healthy tendon tissue progressively surrounds the zone of disorganization in an increasingly consistent way. In the MRI fluid-sensitive sequences (T2 and STIR), the area of healthy tendon tissue has a hypoechoic appearance. On the contrary, the area of tissue disorganization appears as an area of hyperintensity. This process, from an iconographic point of view, justifies the “doughnut metaphor” [26,27,28]. Indeed, as the reparative processes progress, MRI images increasingly resemble a doughnut.

**Figure 2 ijerph-20-05985-f002:**
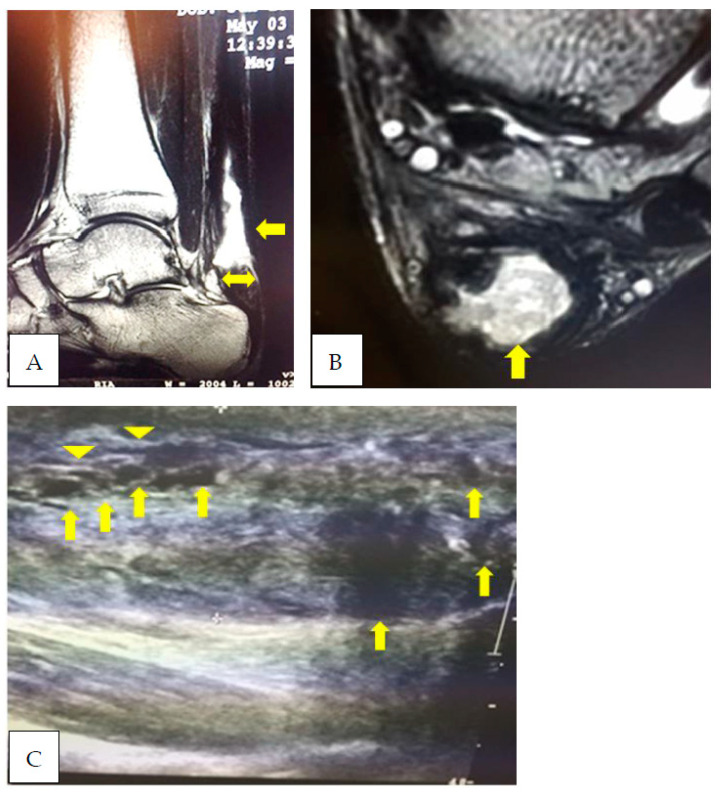
Five months after surgery the sagittal T2 (**A**) and coronal STIR (**B**) images show an important zone of signal hyperintensity due to the disorganization area corresponding to the zone subjected to tenorraphy (arrow). This zone of disorganization is ~72% of the entire tendon cross-section area (CSA). In sagittal images (**A**), the tendon appears thickened in its antero-posterior aspect (24 mm) and the tendon’s anterior margin bulges anteriorly above the calcaneal corner (double arrow). The US examination (**C**) shows several irregular hypoechoic areas (arrows) in a mix of increased and decreased echogenic areas. Furthermore, from the US image, it is possible to note that the tendon fibers are thin and show an irregular delineation (arrowheads).

**Figure 3 ijerph-20-05985-f003:**
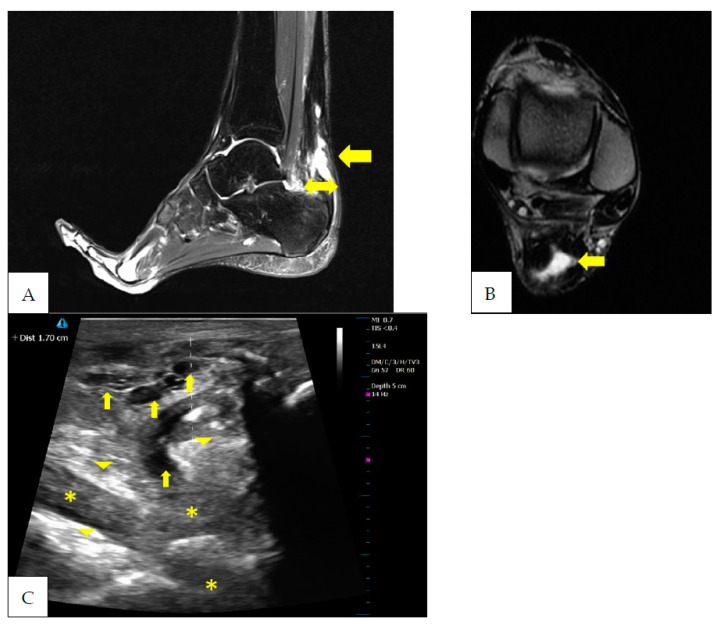
Seven months after surgery the margins of the Achilles tendon are better visualized. The sagittal (**A**) and coronal (**B**) STIR images show a reduction in the hyperintensity area equal to ~30% of the tendon CSA (arrow). Furthermore, is possible to note a centralization of the hyperintensity area. Moreover, it is possible to note that the periphery of the healing tendon shows a normal low-intensity signal. In sagittal images (**A**), the tendon appears thickened in its antero-posterior aspect (22 mm) and the tendon anterior margin bulges less anteriorly above the calcaneal corner (double arrow). The US assessment (**C**) shows a reduction in the hypoechoic areas (arrows). However, it is possible to note the persistence of a combination of increased (arrowheads) and decreased (asterisk) echogenic areas.

**Figure 4 ijerph-20-05985-f004:**
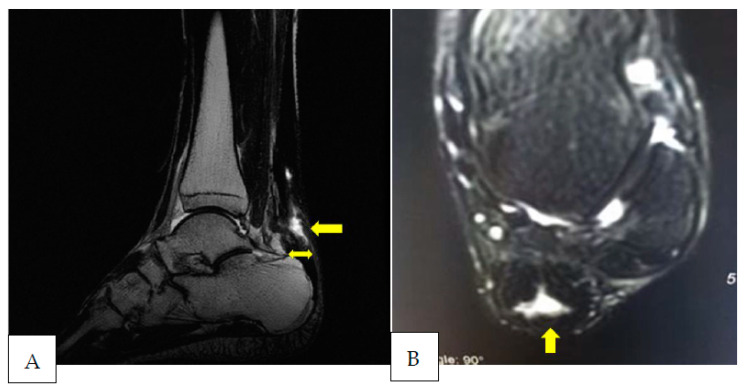
Nine months after surgery the sagittal T2 (**A**) and coronal STIR (**B**) images show a further reduction in the hyperintensity signal area (equal to ~23% of the tendon CSA) and its progressive centralization (arrow). In sagittal images (**A**), the tendon appears even thinner in its antero-posterior aspect (20 mm). The tendon bulging over the calcaneus is additionally reduced (double harrow). The US examination (**C**) shows a reduction in hypoechoic areas (arrows) and the persistence of only a few hyperechoic spots (arrowheads).

**Figure 5 ijerph-20-05985-f005:**
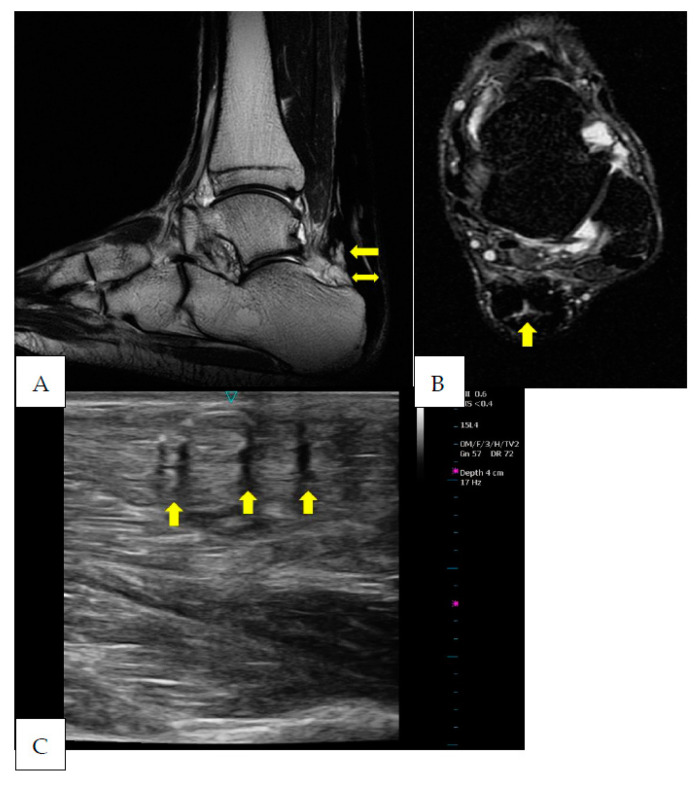
Twelve months after surgery the T2 sagittal (**A**) and coronal STIR (**B**) images show that the hyperintensity area was reduced to ~2% of the tendon CSA. In sagittal images (**A**), the tendon appears even thinner in its antero-posterior aspect (18 mm) and the tendon bulging over the calcaneus is barely visible. The US image (**C**) shows a total disappearance of the hypoechoic areas and hyperechoic spots. Only hypoanechoic areas corresponding to the sutures (arrows) are noted.

**Figure 6 ijerph-20-05985-f006:**
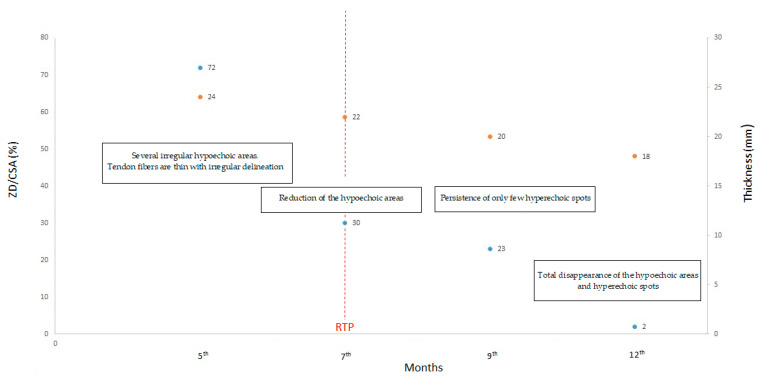
Summary graph of the main data from the MRI and US examinations. In ordinate: on the left main axis, the percentage ratio (blue circles) between zone of degeneration (ZD) and cross-section area of the tendon (CSA). In ordinate: on the secondary right axis, the antero-posterior thickness of the tendon expressed in millimeters (orange circles). In abscissa: the post-surgical months. In the boxes: a summary of the main findings of the US assessment. The dashed red line indicates the return-to-play (RTP) period.

## Data Availability

The data presented in this work are available upon request from the corresponding author.

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
