# Peer review of "Achilles Tendon Repair after Tenorraphy Imaging and the Doughnut Metaphor"

_ijerph, 2023, doi:10.3390/ijerph20115985_

Round 1

Reviewer 1 Report

The authors have presented an interesting paper that is of clinical interest when making RTP decisions facing tendon ruptures.

I have a few comments regarding the elaboration of the manuscript.

Even though the information is contained, the reporting of the manuscript should follow guidelines according to the type of study. This is to ensure every element of the manuscript is included.

The manuscript is missing a "Methods" and a "Results" section, properly structured. Reporting these sections will allow a better understanding of the Discussion and Conclusion.

Please select the proper EQUATOR Network reporting guideline and follow it to improve manuscript reporting.

I don't understand why do you state in Conclusions "US may represent a very important tool during the RTP decision-making process" but argue extensively in Discussion that tissue repair is heterogenous. If image is not really associated with funcionality and players can return to play despite not having imaging techniques to support that decision, I beleive it is bold to state that imaging is a "very important" tool. I'd rather consider it useful or complementary, in this case.

Author Response

REVIEWER 1

Comments and Suggestions for Authors

The authors have presented an interesting paper that is of clinical interest when making RTP decisions facing tendon ruptures.

I have a few comments regarding the elaboration of the manuscript.

Even though the information is contained, the reporting of the manuscript should follow guidelines according to the type of study. This is to ensure every element of the manuscript is included.

Reply

We reset the manuscript following the EQUATOR Network reporting guidelines regarding the “ Case Reports”

The manuscript is missing a "Methods" and a "Results" section, properly structured. Reporting these sections will allow a better understanding of the Discussion and Conclusion.

Reply

Since, as requested we reset the manuscript following the EQUATOR Network reporting guidelines regarding the “ Case Reports” (which don’t include the sections “Methods” and “Results”) the manuscript includes the following sections: “Introduction” , “Case presentation”, “Discussion”, “Limitation of the study and future implications” and  “Conclusions”.

Please select the proper EQUATOR Network reporting guideline and follow it to improve manuscript reporting.

Reply

We reset the manuscript following the EQUATOR Network reporting guidelines regarding the “ Case Reports”

I don't understand why do you state in Conclusions "US may represent a very important tool during the RTP decision-making process" but argue extensively in Discussion that tissue repair is heterogenous. If image is not really associated with funcionality and players can return to play despite not having imaging techniques to support that decision, I beleive it is bold to state that imaging is a "very important" tool. I'd rather consider it useful or complementary, in this case.

Reply

We changed the sentence “a very important tool during..” in “ an useful tool during …”

Reviewer 2 Report

Thank you for this helpful contribution. I applaud the effort of promoting studies for the investigation of tendon surgical procedures. I appreciate their methods including study design and data analysis. I write some comments below that could benefit the article.

Figures. Information is shown in duplicate. If it appears in the figure, delete the information from the text.

This study is limited because it is a clinical case without sample size, I invite the authors to continue researching in the same research. No scientific conclusion can be reached without a clinical trial. I recommend authors follow the consort statement for randomized clinical trials.

References. It is advisable to reference articles published in scientific journals.

Thanks again!

Author Response

REVIEWER 2

Comments and Suggestions for Authors

Thank you for this helpful contribution. I applaud the effort of promoting studies for the investigation of tendon surgical procedures. I appreciate their methods including study design and data analysis. I write some comments below that could benefit the article.

Figures. Information is shown in duplicate. If it appears in the figure, delete the information from the text.

Reply.

We have checked that the information appearing in the caption of the figures is not duplicated in the text.

This study is limited because it is a clinical case without sample size, I invite the authors to continue researching in the same research. No scientific conclusion can be reached without a clinical trial. I recommend authors follow the consort statement for randomized clinical trials.

Reply

We totally agree with the reviewer. We have included a "Limitations of the study and future implications" section where we specify what the reviewer said.

References. It is advisable to reference articles published in scientific journals.

Reply

We have checked that all references are in PubMed. We have also replaced the only reference not present in PubMed (i.e. Draghi et al., 1999)

Reviewer 3 Report

Dear Authors, Excellently written case report, however few minor corrections can be found in the PDF.

Author Response

REVIEVER 3

The beginning of the case introduction is not clear in the abstract . However at the end you mentioned “therefore for the athlete”. Please properly introduce the case in the abstract and complete the abstract by your success

Reply

We have rephrased the abstract introducing the case more clearly

Even tough we very commonly know STIR image means STIR (Short Tau Inversion Recovery) images. But we need to mention it at least once for readers sake

Reply

The abbreviation STIR (Short Tau Inversion Recovery)  has been made explicit in the text

At this heading adding a title of case description will improve the readers understanding at same time the guidelines of case study also will be followed

Reply

We changed the old heading (Recovery of tendon functionality according to the reparative processes progress: a practical example) in “Case presentation”.

Victoria Institute of Sport Assessment –Achilles. I think you mean this for VISA-A score – Please mention the abbreviation.

Reply

We have mentioned the abbreviation as required

Limitation of your study and future implications will further boost the next generation researches to kick start a new research. Please mention them

Reply

We have introduced a new section for the Limitation of the study and future implications

Reviewer 4 Report

For authors, 

This paper provides interesting findings and practical applications regarding the US and MRI images technique for tendon tissue injury recovery. The novelty of this case study seems to be relevant for the scientific community in sports..  Even so, I will expose the limitations that  were identified and should be revised by the authors, being:

1st -  I recommend for authors to read the Lin et al. (2004) entitled : “Biomechanics of tendon and injury repair, mainly at the section 4 tendon injury and repair  to reinforce some aspects of the introduction and discussion sections.

2nd – Most sentences are too long, making it difficult for the reader understanding. I suggest you review this throughout the manuscript.

3rd  – The authors did not include the reliability values of the measurements using US and MRI techniques.

4th – Figures should be revised . I suggest authors to expose some graphs related to the evolution of the football player.  

5th – The authors should include during the discussion section the main limitations of this study

Specific comments

Abstract: The keywords should be different that those presented in the abstract text.

Introduction: 

Line 48 - The abbreviation STIR was not quoted before. Please, correct.

Line 69-70 -  The abbreviation C was not quoted before. Please rewrite this sentence.

Line 102- 103 – As a case study, it is interesting to reveal the history of training, competition and tendon injuries of this specific football player.  Other data are also relevant, body mass and  height

Results:

It is so difficult to see the results by images. Could authors bring for readers some graphs showing the evolution of the tendon regenerating  process?

Discussion:

A brief limitation of this study should be presented by the authors.

Author Response

REVIEVER 4

For authors, 

This paper provides interesting findings and practical applications regarding the US and MRI images technique for tendon tissue injury recovery. The novelty of this case study seems to be relevant for the scientific community in sports..  Even so, I will expose the limitations that  were identified and should be revised by the authors, being:

1st -  I recommend for authors to read the Lin et al. (2004) entitled : “Biomechanics of tendon and injury repair, mainly at the section 4 tendon injury and repair  to reinforce some aspects of the introduction and discussion sections.

 Reply

We have consulted the indicated article (the Lin et al. 2004) and included some concepts present in the article both in the introduction and in the discussion

2nd – Most sentences are too long, making it difficult for the reader understanding. I suggest you review this throughout the manuscript.

Reply

We checked and modified the too long sentences throughout the manuscript as requested

3rd  – The authors did not include the reliability values of the measurements using US and MRI techniques.

Reply

We included the reliability values of the US and MRI measurements

4th – Figures should be revised . I suggest authors to expose some graphs related to the evolution of the football player.  

 Reply

We added a graph showing the tendon regenerating process.

5th – The authors should include during the discussion section the main limitations of this study

  Reply

We added a section concerning the limitations of the study.

Specific comments

Abstract: The keywords should be different that those presented in the abstract text.

 Reply

We changed the keywords

Introduction: 

Line 48 - The abbreviation STIR was not quoted before. Please, correct.

Reply

The abbreviation STIR (Short Tau Inversion Recovery)  has been made explicit in the text

Line 69-70 -  The abbreviation C was not quoted before. Please rewrite this sentence.

Reply

 The abbreviation 14C (Carbon-14, or radiocarbon ) has been made explicit in the text

Line 102- 103 – As a case study, it is interesting to reveal the history of training, competition and tendon injuries of this specific football player.  Other data are also relevant, body mass and  height

 Reply

We added a brief history of training and competition. Furthermore, as request we added the anthropometric data of the subject.

Results:

It is so difficult to see the results by images. Could authors bring for readers some graphs showing the evolution of the tendon regenerating  process?

Reply

We added a graph showing the tendon regenerating process.

Discussion:

A brief limitation of this study should be presented by the authors.

 Reply

We added a section concerning the limitations of the study.

Round 2

Reviewer 2 Report

Congratsss!!

Reviewer 4 Report

Dear Editor, 

Many improvements have been done in the manuscript. The authors followed all corrections proposed. I suggest it for publication.